# Combine and Conquer: A Meta-Analysis on Data Shift and Out-of-Distribution Detection

**Eduardo Dadalto**[1,4]**, Florence Alberge**[2]**, Pierre Duhamel**[1,4]**, Pablo Piantanida**[3,4]
[1]**Laboratoire des Signaux et Systèmes (L2S), Université Paris-Saclay**
[2]**SATIE Laboratory, Université Paris-Saclay, ENS Paris-Saclay, CNRS**
[3]**ILLS - International Laboratory on Learning Systems, Mila - Quebec AI Institute**
[4]**CNRS, CentraleSupélec**

*Reviewed on OpenReview:* `https://openreview.net/forum?id=VGNBUS9TrU`

## Abstract

This paper introduces a universal approach to seamlessly combine out-of-distribution (OOD) detection scores. These scores encompass a wide range of techniques that leverage the self-confidence of deep learning models and the anomalous behavior of features in the latent space. Not surprisingly, combining such a varied population using simple statistics proves inadequate. To overcome this challenge, we propose a quantile normalization to map these scores into p-values, effectively framing the problem into a multi-variate hypothesis test. Then, we combine these tests using established meta-analysis tools, resulting in a more effective detector with consolidated decision boundaries. Furthermore, we create a probabilistic interpretable criterion by mapping the final statistics into a distribution with known parameters. Through empirical investigation, we explore different types of shifts, each exerting varying degrees of impact on data. Our results demonstrate that our approach significantly improves overall robustness and performance across diverse OOD detection scenarios. Notably, our framework is easily extensible for future developments in detection scores and stands as the first to combine decision boundaries in this context. The code and artifacts associated with this work are publicly available[1].

## 1 Introduction

Deploying AI systems in real-world applications is not without its challenges. Although these systems are evaluated in static scenarios, they encounter a dynamic and evolving environment in practice. One of the most pressing issues is preventing and reacting to *data distribution shift* (Quionero-Candela et al., 2009). It occurs when the data distribution used to train an AI model no longer matches the data required to process in test time. It can happen gradually or suddenly and can be caused by various factors, e.g., changes in user behavior or degradation in operating conditions, which can have severe consequences in safety-critical applications (Amodei et al., 2016) such as autonomous vehicle control (Bojarski et al., 2016) and medical diagnosis (Subbaswamy & Saria, 2020). For instance, a predictive model of the Earth's temperature based on historical data may face challenges due to the evolving nature of climate change. Historical patterns and trends may become less reliable indicators of future temperature changes, which could undermine the dangers of climate change unless a mechanism to detect such drifts is in place.

Modern machine learning models can be difficult and expensive to adapt. Even though shifts in distributions can result in significant performance decline, in reality, distributions also undergo harmless shifts (Gemaque et al., 2020). As a result, practitioners should focus on discerning detrimental shifts that harm predictive performance from unimportant shifts that have little impact. This paper explores ways to improve the *detection* of performance-degrading shifts by ensembling existing detectors in an unsupervised manner. Each

---

[1]`https://github.com/edadaltocg/detectors`

detector can be formalized as a test of equivalence of the source distribution (from which training data is sampled) and target distribution (from which real-world data is sampled) through the lens of a predictive model. Our approach is motivated by the fact that different detection algorithms may make trivial mistakes in different parts of the data space without any assumptions on the test data distribution (Birnbaum, 1954). The challenge is to develop a widely applicable method for combining detectors to alleviate individual catastrophic mistakes.

Combine and Conquer draws inspiration from *meta-analysis* (Glass, 1976), which consolidates findings from various statistical hypothesis tests to derive a unified estimate. To the best of our knowledge, this is the first work employing such methodologies in out-of-distribution (OOD) and data distribution shifts detection. Given that the underlying distributions (in-distribution and out distribution) are unknown and only partial information (derived from the training and validating samples) is available about one of the hypotheses, each score may carry relevant information for the decision. Each score induces a distinct probability mass transformation of the same data point, contingent upon the underlying hypothesis under test. Viewed in this light, this multi-score approach can be seen as a means of enhancing the diversity of the scores by aggregating different scoring mechanisms, which is useful for testing different out-of-distribution scenarios. Consequently, this increases the likelihood that at least one will successfully identify the correct hypothesis.

We summarize our **contributions** as follows:

1. We present a simple and convenient ensembling algorithm for combining existing out-of-distribution data detectors, leading to better generalizability by incorporating effects that may not be apparent in individual detectors.

2. A probabilistic interpretable detection criterion is obtained by adjusting the final statistics to align with a distribution characterized by known parameters.

3. A framework to adapt any single example detector to a window-based data shift detector.

We validate our contributions through a comprehensive empirical investigation encompassing classic OOD detection and introduce a benchmark on window-based data distribution shift detection.

## 2 Related Works

**Window-based data shift detection.** This line of work proposes methods for detecting shifts in data distribution using multiple samples. Lipton et al. (2018) presents a technique for detecting prior probability shifts. Rabanser et al. (2019) studies two-sample tests with high dimensional inputs through dimensionality reduction techniques from the input space to a projected space. Cobb & Looveren (2022) explores two sample conditional distributional shift detection based on maximum conditional mean discrepancies to segment relevant contexts in which data drift is diminishing. These studies, along with our own, demonstrate detection methods for detecting shifts in windowed data. For a survey on *adapting* models to these shifts, please refer to Gama et al. (2014) and Lange et al. (2022).

**Misclassification detection.** Misclassification detection aims to reject in-distribution samples misclassified in test time with roots in rejection option (Chow, 1957) and uncertainty quantification (Abdar et al., 2021). A natural baseline is the classification model's maximum softmax output (Hendrycks & Gimpel, 2017; Geifman & El-Yaniv, 2017). Other works (Granese et al., 2021), introduced a framework that considers the entire probability vector output to detect misclassifications. Gal & Ghahramani (2016); Lakshminarayanan et al. (2016) are popular approaches for estimating uncertainty from a Bayesian inference perspective. Even though this line of work focuses mainly on detecting problematic in-distribution samples while we focus on distributional drifts, our framework could be extended to it.

**Out-of-distribution detection.** OOD detection is also referred to in the literature as open-set recognition (Geng et al., 2021), one-class novelty detection (Pimentel et al., 2014), or semantic anomaly detection (Wang et al., 2020). Overall, methods are taxonomized into confidence-based Hein et al. (2019); Hendrycks & Gimpel (2017); Liang et al. (2018); Hsu et al. (2020); Liu et al. (2020); Hendrycks et al. (2022); Sun & Li

(2022), which rely on the logits and softmax outputs; feature-based (Sastry & Oore, 2020; Quintanilha et al., 2019; Huang et al., 2021; Zhu et al., 2022; Colombo et al., 2022; Dong et al., 2021; Song et al., 2022; Lin et al., 2021; Djurisic et al., 2023; Lee et al., 2018; Ren et al., 2021; Sun et al., 2022; Darrin et al., 2023), which explores latent representations; mixed feature-logits (Sun et al., 2021; Gomes et al., 2022; Wang et al., 2022; Dadalto et al., 2023; Djurisic et al., 2023); training, likelihood estimation and reconstruction based (Schlegl et al., 2017; Vernekar et al., 2019; Xiao et al., 2020; Ren et al., 2019; Zhang et al., 2021; Kirichenko et al., 2020) methods. We consider these methods to be complementary to our work as they focus on developing single discriminative OOD scores. The authors in Haroush et al. (2022) propose a comparable approach for OOD detection, framing it as a statistical hypothesis testing issue. They aggregate p-values based on statistics obtained from various channels of a single convolutional network in a hierarchical manner. However, this approach is heavily reliant on the architecture of convolutional neural networks and dimension reduction functions. It does not account for the correlation between the test statistics, as highlighted in Section 4.2 therein. Moreover, a recent benchmark (Zhang et al., 2023) shows no evident winner in detecting OOD data. In this paper, we introduce a novel approach that involves combining multiple detectors to enhance performance and mitigate the risk of catastrophic failures when a specific method fails to detect certain types of data.

## 3 Preliminaries and Methodology

This section discusses the methodology for detecting distribution shifts in high dimensional data streams inputted to deep neural networks. We define data stream in Section 3.1, we recall the various types of shifts in Section 3.2, and we formalize single sample and window-based detection in Section 3.3.

### 3.1 Background

Let $\mathcal{X} \subseteq \mathbb{R}^d$ be a continuous feature space, and let $\mathcal{Y} = \{1, \ldots, C\}$ denote the label space related to some task of interest. We denote by $p_{XY}$ and $q_{XY}$ the underlying source and target probability density functions (pdf) associated with the distributions $P$ and $Q$ on $\mathcal{X} \times \mathcal{Y}$, respectively. We assume that a machine learning model $f : \mathcal{X} \to \mathcal{Y}$ is trained on some training set $\mathcal{D}_n = \{(\boldsymbol{x}_1, y_1), \ldots, (\boldsymbol{x}_n, y_n)\} \sim p_{XY}$, which yields a model that, given an input $\boldsymbol{x} \in \mathcal{X}$, outputs a prediction on $\mathcal{Y}$, i.e., $f(\boldsymbol{x}) = \arg\max_{y \in \mathcal{Y}} p_{\hat{Y}|X}(y \mid \boldsymbol{x})$. At test time, an unlabeled sequence of inputs or *data stream* is expected, sampled from the marginal target distribution $q_X$.

**Definition 3.1** (Data stream)**.** A data stream $\mathcal{S}$ is a finite or infinite sequence of not necessarily independent observations typically grouped into *windows* (i.e., sets $\mathcal{W}_j^m = \{x_j, \ldots, x_{j+m-1}\} \sim q_X$) of same size $m$,

$$\mathcal{S} = \{\boldsymbol{x}_1, \ldots, \boldsymbol{x}_m, \ldots\} = \bigcup_{j=1}^{\infty} \mathcal{W}_j^m. \tag{1}$$

### 3.2 Data-Shift

In real-world applications, data streams usually suffer from a well-studied phenomenon known as *data distribution shift*[2] (or data shift for short). Data shift occurs when the test data joint probability distribution differs from the distribution a model expects, i.e., $p_{XY}(\boldsymbol{x}, y) \neq q_{XY}(\boldsymbol{x}, y)$. Due to this mismatch, the model's response may suffer a drop in accuracy. Let $\beta \in [0, 1]$ be a mixture coefficient, we will write the true joint test pdf $q_{XY}$ as a mixture of pdfs $p$ and $v$[3]:

$$q_{XY}(\boldsymbol{x}, y) = (1 - \beta) \cdot p_{XY}(\boldsymbol{x}, y) + \beta \cdot v_{XY}(\boldsymbol{x}, y). \tag{2}$$

Note that when $\beta = 0$, the test distribution matches the training distribution, i.e., there is no shift. Conversely, when $\beta = 1$, we have the largest shift between training and testing environments. In this work, we

---

[2] Also referred to in the literature as data distribution *drift*.
[3] We assume that $v$ is unknown and differs significantly from $p$, i.e., $\frac{1}{2} \int_{\mathcal{X} \times \mathcal{Y}} |p(z) - v(z)| dz \geq \delta$.

focus on detecting when *any* kind of data shift happens between the training and testing distributions and not estimating the mixing parameter $\beta$ or the true pdfs involved.

One could categorize different kinds of shifts that may happen by decomposing a joint pdf into

$$q(X,Y) = \underbrace{Q(Y|X)}_{\text{posterior}}\underbrace{q(X)}_{\text{covariate}} = \underbrace{q(X|Y)}_{\text{likelihood}}\underbrace{Q(Y)}_{\text{prior}}. \tag{3}$$

Briefly, *novelty drift*, also referred to as concept evolution (Masud et al., 2011), is usually attributed to the presence of novel classes or concepts. As a result, the conditional distribution is not adapted anymore, i.e., $P(Y|X) \neq Q(Y|X)$. Naturally, $q(X|Y)$, $Q(Y)$, and $Q(X)$ are allowed to change. *Covariate shift* often happens because the input data comes from a different domain as $q(X)$ changes, e.g., in image recognition, the drawing of concepts are introduced in testing, while the training features are natural pictures only. Finally, a *prior shift* or label shift usually occurs when the test label distribution is biased towards some classes, e.g., the majority of samples in testing come from a class in which the predictive model is less proficient, causing overall accuracy degradation. All of these *data shifts* may have negative impacts on the model. Shifts that do not affect the classifier's performance are referred to *virtual* drifts, e.g., mild corruptions to images may result in $q(X)$ drifting but without affecting $Q(Y|X)$. However, this list is not exhaustive as the principal objective of this paper does not lie in the precise categorization of diverse drift phenomena but rather in the establishment of a more robust detection framework for distinct scenarios.

Considering our objective of enhancing the overall reliability of AI systems in real-world applications, our focus is detecting *any* form of data drift that might result in model deterioration without having access to labeled samples. Consequently, our primary emphasis will be on detecting data drift by examining the discriminative model, as the high dimensionality of the input data poses significant challenges for generative modeling approaches. Through empirical validation, we demonstrate that our strategy can proficiently detect novelty and covariate shifts by measuring detection performance on provoked covariate shift and when introduced novel concepts.

### 3.3 Detection Framework

Predictions on a production AI system can be made sample by sample or window by window in a data stream. Both can be interpreted as a statistical hypothesis test.

#### 3.3.1 Single Example Detection

On a **single example** level (equivalent to OOD detection), let $s : (\boldsymbol{x}, f) \mapsto \mathbb{R}$ be a confidence-aware score function that measures how adapted the input is to the model. A low score indicates the sample is untrustworthy, and a high value indicates otherwise. This score can be simply converted to a binary detector through a threshold $\gamma \in \mathbb{R}$, i.e., $d(\cdot) = \mathbb{1}\left[s(\cdot, f) \leq \gamma\right]$. Finally, the role of the system $(d, f)$ is only to keep a prediction if the input sample $\boldsymbol{x}$ is not rejected by the detector $d$, i.e., if $d(\boldsymbol{x}) = 0$. This setup is identical to novelty, anomaly, or OOD detection. Formally, the null and alternative hypothesis writes:

$$H_0 : (X, \widehat{Y}) \sim p_{XY} \text{ and } H_A : (X, \widehat{Y}) \sim q_{XY}. \tag{4}$$

We assume that the score functions are confidence oriented, i.e., greater values indicate more confidence in prediction. So, we frame the statistical hypothesis test as a *left-tailed test* (Lehmann & Romano, 2005). Even though single-sample detection is adapted for anomaly detection, it is not well adapted for detecting distribution shifts.

#### 3.3.2 Multiple Examples Detection

In a **window based detection** scenario, we make the assumptions that 1.) there are multiple reference samples available, 2.) the instance's class label *are not* available right after prediction, and 3.) the model is not updated. So, given a *reference window* $\mathcal{W}_1^r \sim p_{XY}$ with $r$ samples and test window $\mathcal{W}_2^m = \{\boldsymbol{x}_1', \dots, \boldsymbol{x}_m'\} \sim q_X$ with sample size $m$, our task is to determine whether they are both sampled from the source distribution or,

equivalently, whether $p_{XY}(\boldsymbol{x}, y)$ equals $q_{X\widehat{Y}}(\boldsymbol{x}', \hat{y}')$ where $\hat{y}' = f(\boldsymbol{x}')$. The null and alternative hypothesis of the two-sample test of homogeneity writes:

$$H_0 : p_{XY}(\boldsymbol{x}, y) = q_{X\widehat{Y}}(\boldsymbol{x}', \hat{y}') \text{ and } H_A : p_{XY}(\boldsymbol{x}, y) \neq q_{X\widehat{Y}}(\boldsymbol{x}', \hat{y}'). \tag{5}$$

In this case, the null hypothesis is that the two distributions are identical for all $(\boldsymbol{x}, y)$; the alternative is that they are not identical, which is a two-sided test. As testing this null hypothesis on a continuous and high dimensional space is unfeasible, we will compute a univariate score on each sample of the windows. With a slight abuse of notation let $s(\mathcal{W}^m, f) = \{s(\boldsymbol{x}_1, f), \dots, s(\boldsymbol{x}_m, f)\}$ be a multivariate *proxy variable* to derive a unified large-scale window-based data shift detector. To compute the final window score, we rely on the Kolmogorov-Smirnov (Massey, 1951) two-sample hypothesis test over the proxy variable. The test statistic writes:

$$\text{KS}(\mathcal{W}_1^r, \mathcal{W}_2^m) = \sup_{\delta \in \mathbb{R}} \left| \widehat{F}_2^m(\delta) - \widehat{F}_1^r(\delta) \right|, \tag{6}$$

where $\widehat{F}_1^r$ and $\widehat{F}_2^m$ are the empirical cumulative distribution functions (ecdf) of the scores of each sample of the first and the second widows, respectively, as defined in Equation (7). Finally, the KS statistic is compared to a threshold $\gamma \in \mathbb{R}$ to obtain the window-based binary detector $D(\cdot) = \mathbb{1}[\text{KS}(\cdot, \mathcal{W}_1^r) \leq \gamma]$.

## 4 Main Contribution: Arbitrary Scores Combination

This section explains in detail the core contribution of the paper: an algorithm to effectively combine arbitrary detection score functions from a diverse family of detectors. Section 4.1 discusses why basic statistics may fail to combine OOD detectors and motivates a more principled approach based on *meta-analysis* (Glass, 1976), a statistical technique that combines the results of multiple studies to produce a single overall estimate.

The first step of our Combine and Conquer algorithm is to transform the individual scores into p-values through a quantile normalization (Conover & Iman, 1981) (cf. Section 4.2). Then, with multiple detectors, the p-values can be combined using a p-value combination method (cf. Section 4.3). Finally, we introduce an additional statistical treatment, since the p-values of the multiple tests over the same sample are often correlated (cf. Section 4.4).

### 4.1 Simple Normalization for Score Aggregation May Fall Short

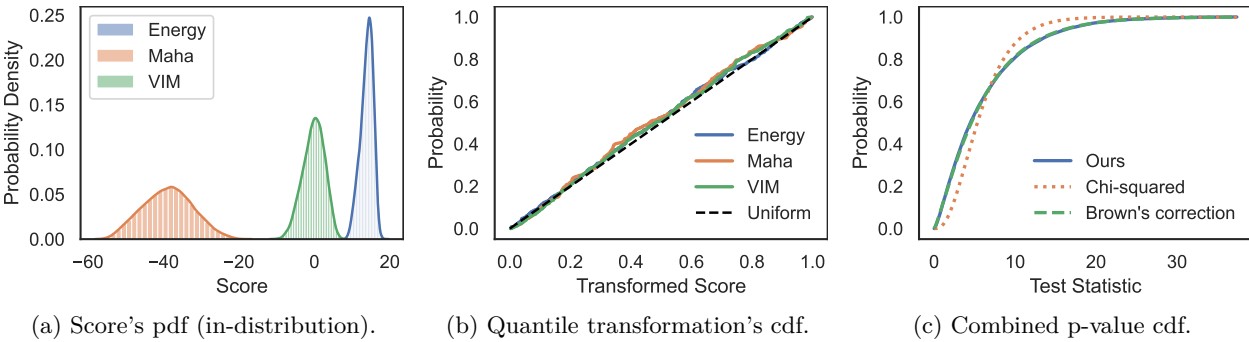

(a) Score's pdf (in-distribution).  (b) Quantile transformation's cdf.  (c) Combined p-value cdf.

Figure 1: Illustration of the three steps of the Combine and Conquer algorithm. This example shows three disparate score functions evaluated on in-distribution data. Our main experiments combine 14 scores.

Common approaches for combining different detection scores often revolve around calculating a mean (de Carvalho, 2016) of the scores while incorporating certain assumptions. These assumptions typically entail considerations such as whether all scores should contribute equally to the final composite score, or if a weighted sum should be computed, assigning greater importance to select methods. Additionally, there's the question of whether outlier score values should be favored over others, and whether a more conservative or permissive approach should be adopted in score combination. For instance, using the product of available scores could

yield a low combined score if any individual scores are low, while selecting the minimum or maximum value among all anomaly scores can influence the method to be more conservative or permissive. While these combination methods are all viable, their effectiveness heavily relies on the distributional characteristics of the involved scores. Given that the choice of aggregation method hinges on the data's characteristics, it's pertinent to delve into the unique attributes of OOD detection scores.

One inherent limitation in OOD detection is the absence of access to a sufficiently representative dataset of outlier data, which poses challenges for techniques like *metalearning* Opitz & Maclin (1999) and other supervised ensembling methods that require data to train a meta-model. Additionally, detection scores often exhibit distinct distribution shapes with varying moments, as illustrated in Figure 1a. To mitigate some of these effects, several simple statistical approaches are commonly employed. One such method is normal standardization or *z-score normalization*, where each individual score random variable $S_i = s_i(X, f)$ is transformed into a standard score $Z_i = (S_i - \bar{S}_i)/\sigma_{S_i}$, where $Z_i$ represents the distance between the raw score and the population mean in units of the standard deviation $\sigma_{S_i}$. While this approach corrects for the first two moments of the distributions, it does not account for skewness, kurtosis, or multimodality. Another frequently used normalization technique is *min-max scaling*, which involves transforming scores to fall within the range of zero and one using statistics $Z_i = (S_i - \min S_i)/(\max S_i - \min S_i)$. However, min-max scaling fails to address many other characteristics and does not provide control over the resulting distribution's moments, making the task of combining scores more challenging. To tackle this issue, we emphasize the importance of pre-processing the scores using *quantile normalization* instead.

## 4.2 Quantile Normalization: Managing Disparate Score Distributions

Each detector's score r.v. $S_i = s_i(X, f)$ follows very different distributions depending on the model's architecture, the dataset it was trained on, and the score function $s_i$. To combine them effectively, we propose first to apply a quantile normalization (Bolstad et al., 2003), which exhibits interesting statistical properties (Gallón et al., 2013). Let $S_i : \Omega \mapsto \mathbb{R}$ be a continuous univariate r.v. captured by a cumulative density function (cdf) $F_i(\delta) = \Pr(S_i \leq \delta)$ for $i \in \{1, \ldots, k\}$ and $\delta \in \mathbb{R}$. Its *empirical* cdf $\widehat{F}_i : \mathbb{R} \mapsto [0, 1]$ is defined by

$$\widehat{F}_i^r(\delta) = \frac{1}{r} \sum_{i=1}^{r} \mathbb{1}\left[S_i \leq \delta\right], \tag{7}$$

which converges almost surely to the true cdf for every $\delta$ by the Dvoretzky–Kiefer–Wolfowitz–Massart inequality (Massart, 1990). We are going to estimate this function using a subsample of size $r$ of the training or validation set if available. The resulting r.v. is uniformly distributed in the interval $[0, 1]$. As a result, for each detector $i$ and sample $\boldsymbol{x}$, we can obtain a p-value:

$$\mathrm{p}_i(\boldsymbol{x}) = P_{H_0}\left(S_i \leq s_i(\boldsymbol{x}, f)\right) = \Pr\left(S_i \leq s_i(\boldsymbol{x}, f) \mid H_0\right) \approx \widehat{F}_i^r\left(s_i(\boldsymbol{x}, f)\right). \tag{8}$$

A decision is made by comparing the p-value to a desired significance level $\alpha$. If $\mathrm{p} < \alpha$, then the null hypothesis $H_0$ is rejected, and the sample is considered OOD. Even though we derived everything for the single example detection case, this formulation can be extended to the window-based scenario.

## 4.3 Combining Multiple P-Values

Our objective is to aggregate a set of $k \geq 2$ scores (or p-values) so that their synthesis exhibits better properties, such as improved robustness or detection performance, by consolidating each method's decision boundaries. Unfortunately, since $q$ is unknown and $p$ is hard to estimate, designing an optimal test is unfeasible according to Neyman–Pearson's Fundamental Lemma (Lehmann & Romano, 2005). However, there are several possible empirical combination methods, such as Tippett (1931) $\min_i \mathrm{p}_i$, Neyman & Pearson (1933) $2 \sum_i^k \ln(1 - \mathrm{p}_i)$, Wilkinson (1951) $\max_i \mathrm{p}_i$, Edgington (1972) $1/k \sum_{i=1}^{k} \mathrm{p}_i$, and Simes (1986) $\min_i \frac{k}{i} \mathrm{p}_i$ for sorted p-values. We are going to explore in detail the Fisher's method (Fisher, 1925; Mosteller & Fisher, 1948) in the main manuscript, also referred to as the chi-squared method, and Stouffer's method (Stouffer et al., 1949) in the appendix Appendix A.1, as they exhibit good properties that will be explored in the following.

If the p-values are the independent realizations of a uniform distribution, i.e., for in-distribution data, $-2\sum_{i=1}^{k} \ln \mathrm{p}_i \sim \chi_{2k}^2$ follows a chi-squared distribution with $2k$ degrees of freedom. Finally, for a test input $\boldsymbol{x}$, Fisher's detector score function can be defined as

$$s_F(\boldsymbol{x}, f) = -2 \sum_{i=1}^{k} \ln \widehat{F}_i(s_i(\boldsymbol{x}, f)). \tag{9}$$

Fisher's test has interesting qualitative properties, such as sensitivity to the smallest p-value, and it is generally more appropriate for combining positive-valued data (Heard & Rubin-Delanchy, 2017) with matches the properties of most OOD scores.

### 4.4 Correcting for Correlated P-Values

It should be noted that Fisher's method depends on the assumption of independence and uniform distribution of the p-values. However, the p-values for the same input sample are not independent. An in-distribution data point (one with a high score) is likely to receive high scores across various useful score functions. Therefore, if one score function produces a high score for a particular point, it's reasonable to expect that a second score function will also assign a high score to the same data point. Brown (1975) proposes correcting the Fisher statistics for correlation by modeling the r.v $s_F(\cdot)$ using a scaled chi-squared distribution, i.e.,

$$s_F(\cdot) \sim c\chi^2(k'), \quad \text{with} \quad c = \mathrm{Var}(S_F)/(2\mathbb{E}[S_F]) \quad \text{and} \quad k' = 2(\mathbb{E}[S_F])^2/\mathrm{Var}(S_F). \tag{10}$$

With this simple trick, we approach more interpretable results, as we know in advance the distribution followed by the in-distribution data under our combined score. As such, we can leverage calibrated confidence values given by the true cdf and leverage more powerful single-sample statistical tests for window-based data shift detection.

**Remark 1.** *Commonly, a score's binary detection threshold $\gamma$ is set based on a certain quantile of the score's value on an in-distribution validation set. Usually, this value is set to have 95% of entities correctly classified. By combining p-values with Fisher's method and correcting for correlation with Brown's method, we have that the detection threshold is given by $\gamma = F_{c\chi^2(k')}^{-1}(\alpha)$.*

**Remark 2.** *Given that Brown's method involves only linear scaling, it does not lead to any reranking of the scores. Consequently, any evaluation metric used for detection (e.g., AUROC) computed with this method will yield results identical to those obtained with the original Fisher's method statistic. Nevertheless, the benefit of employing Brown's correction lies in the calibration of scores based on a known underlying probability data distribution, as depicted in Figure 1c. This calibration enhances the interpretability of the results.*

Algorithm 1 summarizes the offline steps of Combine and Conquer. Finally, at test time, the aggregated binary detection function for an input sample $\boldsymbol{x}$ writes for a given TPR desired performance $\alpha \in [0,1]$:

$$d(\boldsymbol{x}) = \mathbb{1}\left[ F_{c\chi^2(k')}\left( -2\sum_{i=1}^{k} \ln \widehat{F}_i(s_i(\boldsymbol{x}, f)) \right) \le \alpha \right] = \begin{cases} 1 & \text{shift detected,} \\ 0 & \text{no shift detected.} \end{cases} \tag{11}$$

## 5 Experimental Setup

In this section, we present and detail the experimental setup from a conceptual point of view. For all our main experiments, we set as *in-distribution* dataset *ImageNet-1K* (=ILSVRC2012; Deng et al., 2009) on ResNet (He et al., 2016) and Vision Transformers (Dosovitskiy et al., 2021) models. ImageNet is a large-scale image classification dataset containing 14 million annotated images for training and 50,000 annotated images for testing. It contains 1,000 different categories, one per image. Our experiments encompass a full-spectrum setting on i.) classic OOD via single example detection (Section 5.1), ii.) novelty shift via independent window-based detection (Section 5.2; Par. 1), iii.) covariate shift via independent window-based detection (Section 5.2; Par. 2), and iv.) sequential shift detection via sequential window-based detection (Section 5.3).

---

**Algorithm 1** Offline preparation algorithm for combining multiple detectors for OOD detection.

---

**Require:** Classifier $f$, in-distribution held-out data set $\mathcal{D}_r = \{\boldsymbol{x}_1, \ldots, \boldsymbol{x}_r\}$, and $k \geq 2$ detection score functions denoted by $s_1, \ldots s_k$.

---

$S \leftarrow \mathbf{0}_{r \times k}$        $\triangleright$ Initialize empty $r \times k$ matrix
**for** $\boldsymbol{x}_i \in \mathcal{D}_r$ **do**        $\triangleright$ Fill the matrix with in-distribution scores
    **for** $j \in \{1, \ldots, k\}$ **do**
        $S_{i,j} \leftarrow s_j(\boldsymbol{x}_i)$
    **end for**
**end for**
**for** $j \in \{1, \ldots, k\}$ **do**        $\triangleright$ Define the empirical cdfs to compute p-values
    $\widehat{F}_j(\cdot) \leftarrow 1/r \sum_{i=1}^r \mathbb{1}[S_{i,j} \leq \cdot\,]$
**end for**
    $\triangleright$ The following steps are for the Fisher-Brown method. They can be easily adapted to other methods
**for** $i \in \{1, \ldots, r\}$ **do**
    $\mathrm{p}_i \leftarrow -2 \sum_{j=1}^k \ln \widehat{F}_j(S_{i,j})$
**end for**
$\mu \leftarrow 1/r \sum_{i=1}^r \mathrm{p}_i, \quad \sigma^2 \leftarrow 1/r \sum_{i=1}^r (\mathrm{p}_i - \mu)^2$
$c \leftarrow \sigma^2/(2\mu), \quad k' \leftarrow 2\mu^2/\sigma^2$
**return** $\widehat{F}_1, \ldots, \widehat{F}_k, c, k'$

---

## 5.1 Classic Out-of-Distribution Detection

OOD detection benchmarks are created by appending in the same set in-distribution data (from the testing dataset) and novelty from different datasets (Hendrycks & Gimpel, 2017). We evaluate the performance of the detectors by mixing the 50,000 testing samples from ImageNet with the curated **datasets** from Bitterwolf et al. (2023) that contain a clean subset without any semantic overlap with ImageNet of far-OOD datasets: SSB-Easy (Vaze et al., 2022) (farthest novelty from ImageNet-21K ranked by the total path distance between their nodes in the semantic trees of the two datasets), OpenImage-O (OI-O) (Wang et al., 2022) (images belonging to novel classes from the OpenImage-v3 dataset (Krasin et al., 2017)), Places (Zhou et al., 2017) (images of 365 natural scenes categories, e.g., patio, courtyard, swamp, etc.), iNaturalist (Horn et al., 2017) (samples with concepts from 110 plant classes different from ImageNet-1K ones), and Textures (Cimpoi et al., 2014) (collection of textural pattern images observed in nature); and the near-OOD datasets: SSB-Hard (Vaze et al., 2022) (closes novelty from ImageNet-21K ranked by the total path distance between their nodes in the semantic trees of the two datasets), Species (Hendrycks et al., 2022) (plant species sourced from (Horn et al., 2017) not belonging to ImageNet-1K or ImageNet-21K), and finally NINCO (Bitterwolf et al., 2023) (images from 64 categories manually curated from several popular datasets).

For the **evaluation metrics**, we consider the popular Area Under the Receiver Operating Characteristic curve (AUROC), which measures how well the OOD score distinguishes between in-distribution and out-of-distribution data in a threshold-independent manner (higher is better). The ROC curve is constructed by plotting the true positive rate (TPR) against the false positive rate (FPR) at various threshold values. More rigorously, the AUROC corresponds to the probability that a randomly drawn in-distribution sample has a higher score than a randomly drawn OOD sample for a confidence-based score.

For the **baselines**, we consider the following post-hoc detection methods (14 in total): MSP (Hendrycks & Gimpel, 2017), Energy (Liu et al., 2020), Mahalanobis or Maha for short (Lee et al., 2018), Igeood (Gomes et al., 2022), MaxCos (Techapanurak et al., 2020), ReAct (Sun et al., 2021), ODIN (Liang et al., 2018), DICE (Sun & Li, 2022), VIM (Wang et al., 2022), KL-M (Hendrycks et al., 2022), Doctor (Granese et al., 2021), RMD (Ren et al., 2021), KNN (Sun et al., 2022), GradN (Huang et al., 2021). We followed the hyperparameter selection procedure suggested in the original papers when needed. New methods can be easily integrated into our universal framework and should improve the robustness and, potentially, the

performance of the group detector. In Section 6, we delve into the empirical findings and examine whether an optimal subset of detectors exists that enhances detection performance.

## 5.2 Independent Window-Based Detection

**Novelty shift.** To simulate a novelty shift at test time, we fabricate fully ID windows and corrupted windows formed by a mixture of ID and OOD data from the OpenImage-O (OI-O) (Wang et al., 2022) dataset with mixing parameter $\beta$ as defined in Equation (2). As a result, the model will encounter windows that contain novelty from natural images. The objective of the detectors is to classify each test window as being corrupted or not in order to secure the predictor. To do so, each test window is compared to a fixed reference window of size $r = 1000$ extracted from a clean validation set. We ran experiments with $\beta \in [0, 1]$ and with window sizes $|\mathcal{W}| \in \{1, \ldots, 1000\}$. We use the KS two sample test described in Section 3.3 as window-based test statistics. Evaluation metrics and baselines are the same as described in Section 5.1. Figure 2 shows the Fisher's test statistic computed on windows with different mixture amounts and sizes. Figure 2a shows the distribution of the test statistics for different mixture values from $\beta = 0$ (fully ID window) to $\beta = 1$ (fully OOD window). Figure 2b displays how the distribution of the test statistic changes from flatter to peaky as we increase the window size in the simulations (better seen in color). Finally, Figure 2c demonstrates how the detection performance is affected by window size increase and mixture coefficient. As expected, note an AUROC of 0.5 for the case with $\beta = 0$. With a window size as low as 8, we can perfectly distinguish fully corrupted windows from normal ones. Similar qualitative behavior is observed for all detectors.

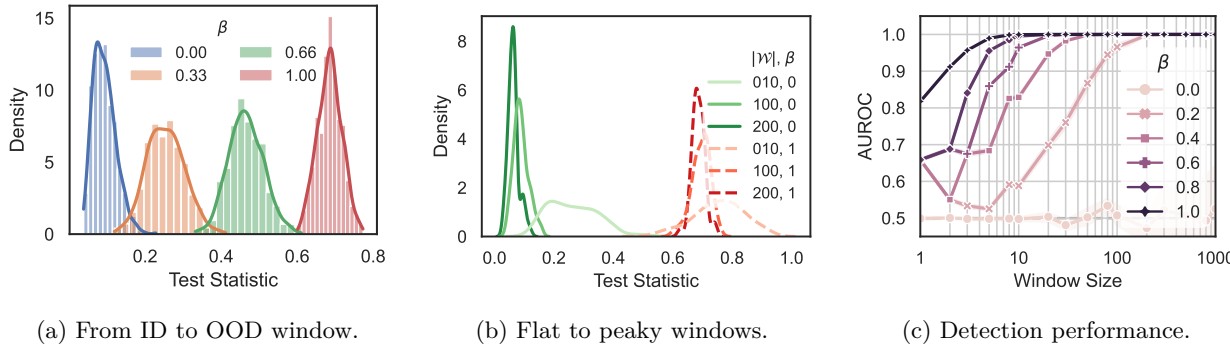

| (a) From ID to OOD window. | (b) Flat to peaky windows. | (c) Detection performance. |

Figure 2: Test statistic distributional behavior and detection performance as a function of the novelty shift intensity and window size. Experiments ran for Fisher's method on a ResNet-50.

**Covariate shift.** To simulate a covariate shift at test time, we ran experiments with the ImageNet-R (IN-R) (Hendrycks et al., 2021) dataset. This dataset contains images from different domains than natural objects for 200 ID classes of ImageNet. The shifted domains are, for instance, cartoons, graffiti, origami, paintings, plastic objects, tattoos, etc.. Similarly to the novelty drift setup described in the previous paragraph, we suppose that the windows arrive independently. We use the same reference window to compute metrics and vary the mix parameter and window size in the same way. Figure 8 is analogous to Figure 2 and shows the behavior of the combined p-values for detecting covariate shift in windows of a data stream. We draw similar qualitative observations from it. Table 1 display the accuracy of each model studied on the new domain. We can see that the drift is severe without masking only the classes present on IN-R, with a top-1 accuracy of around 1% only. However, as we compute the top accuracy only on the 200 classes by

Table 1: Top-1 accuracies in percentage.

| Model | Train | Val. | IN-R | IN-R (m) |
|---|---|---|---|---|
| RN-50 | 87.5 | 76.1 | 1.33 | 36.2 |
| RN-101 | 90.0 | 77.4 | 1.67 | 39.3 |
| RN-152 | 90.2 | 78.3 | 0.67 | 41.4 |
| ViT-S-16 | 88.0 | 81.4 | 1.33 | 46.0 |
| ViT-B-16 | 90.5 | 84.5 | 3.33 | 56.8 |
| ViT-L-16 | 92.3 | 85.8 | 1.67 | 64.3 |

(m)asking the other 800, we can observe an amelioration in performance. In our experiments, we simulate the more realistic and challenging scenario by supposing this mask is unavailable.

### 5.3 Sequential Drift Detection

Unlike the independent window-based detection setting introduced in Section 5.2, in this setup, we implement a sliding window of size 64 with a stride of one, so that the resulting windows contain overlapping data samples. We assume that the samples arrive sequentially and that the labels are unavailable to compute the true accuracy of the model on the current or past test windows. The *objective* is to measure how well the moving average of the detection score will correlate with the moving accuracy of the model. By having a high correlation with accuracy, a machine learning practitioner can use the values of the score as an indicator if the system is suffering from any degrading data distribution shift.

To simulate a progressive sequential drift in a data strem, we ran experiments with the corrupted ImageNet (IN-C) (Hendrycks & Dietterich, 2019) dataset. This dataset contains corrupted versions of ImageNet test data by introducing image pre-processing functions, such as adding noise, changing brightness, pixelating, compressing, etc. The intensity of the drift increases over time from intensity 0 (training warmup set and part of the validation set without corruptions) to 5. Figure 3 illustrates the monitoring pipeline with the moving accuracy on the left y-axis and the score's moving average on the right y-axis. The score's moving average can effectively follow the accuracy (hidden variable). The dashed lines are the timestamps in which the drift intensity progressively increases.

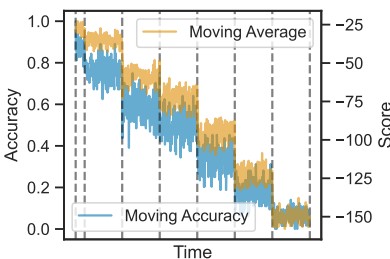

Figure 3: Data stream monitoring with correlation $\rho = 0.98$

## 6 Results and Discussion

**Out-of-distribution detection.** Table 2 displays the experimental result on classic OOD detection for a ResNet-50 model on the setup described in Section 5.1. Fisher's method achieves state-of-the-art results on average AUROC, surpassing the previous SOTA by 1.4% (MaxCos). Also, the other six standard p-value combination strategies also achieve great results, validating our proposed meta-framework of Section 4. Similar FPR and other architectures tables are available in the Appendix A. Apart from achieving overall great performance capabilities, the most compelling observed property is the robustness compared to individual detection metrics. Figure 4 shows the ranking per dataset and on average for selected methods. We can observe that even though several detectors achieve top-1 performance in a few cases, there are several datasets in which they underperform, sometimes catastrophically. This is not true for the group methods, which can effectively combine the existing detectors to obtain a final score that successfully combines the multiple decision regions. For instance, Combine and Conquer with Fisher/Brown keeps top-4 performance in all cases on the ResNet-50 ImageNet benchmark and Stouffer/Hartung is top-5 in all cases.

**Standard approaches for scores combination.** Table 3 shows the performance of simple normalization statistics (min-max scaling and standard normalization) combined with simple score combination methods (mean, min, max). Note that the quantile normalization range is bounded, which helps the interpretability of the scores in a model monitoring pipeline. The range for min-max scaling is not $[0, 1]$ in this case because the statistics are computed on a held-out validation set. We also observe better performance overall for the Combine and Conquer methodology.

**Independent window-based detection.** Figure 5 displays results on novelty shift detection. Figure 5a) shows the detectors' performance with the window size, showcasing a small edge in performance for Vim, Fisher's, and Stouffer's methods. Figure 5b displays the impact of the mixture parameter. Figure 5c shows that model size mildly impacts detection performance, with registered improvements for ResNet-152 and ResNet-101 over ResNet-50 on Fisher's method. The confidence interval bounds are computed over 10 different seeds and are quite narrow for all methods. Similar observations are drawn in the covariate shift

Table 2: Numerical results regarding AUROC (values in percentage) comparing p-value combination methods against literature for a ResNet-50 model trained on ImageNet. The left-hand side shows results on out-of-distribution detection, and the right-hand side shows results on novelty (OI-O) and covariate (IN-R) shift detection with $|\mathcal{W}| = 3$ and $\beta = 1$. We recall the basic combination methods (with their equations): Edgington ($1/k \sum_{i=1}^{k} p_i$), Pearson ($2 \sum_{i}^{k} \ln(1 - p_i)$), Simes ($\min_i \frac{k}{i} p_i$ for sorted p-values), Tippet ($\min_i p_i$), and Wilkinson ($\max_i p_i$).

| | | Out-of-Distribution Detection | | | | | | | | Data Shift Detection | |
| Method | Avg. | SSB-H | NINCO | Spec. | SSB-E | OI-O | Places | iNat. | Text. | IN-R | OI-O |
|---|---|---|---|---|---|---|---|---|---|---|---|
| Fisher/Brown | **89.8** | 75.8 | 84.3 | 88.7 | 91.0 | **93.0** | 93.1 | 95.9 | 96.4 | **94.3** (0.2) | **95.7** (0.4) |
| Stouffer/Hartung | 89.6 | 75.5 | 84.6 | 89.0 | 90.9 | 92.8 | 92.7 | 95.8 | 95.5 | 92.8 (0.2) | 95.5 (0.4) |
| Edgington | 89.3 | 75.2 | 84.6 | 89.0 | **91.0** | 92.5 | 92.1 | 95.5 | 94.4 | 92.5 (0.2) | 95.3 (0.3) |
| Pearson | 89.2 | 74.6 | **84.9** | **89.4** | 90.9 | 92.4 | 91.8 | 95.5 | 94.1 | 92.2 (0.3) | 93.9 (0.4) |
| Simes | 89.2 | 75.0 | 83.0 | 87.6 | 89.5 | 92.3 | 93.1 | 95.7 | 97.0 | 83.6 (0.5) | 86.6 (0.7) |
| Tippet | 88.5 | 74.8 | 80.9 | 86.7 | 87.3 | 91.7 | 93.5 | 95.9 | 97.2 | 82.0 (1.0) | 81.5 (0.7) |
| Wilkinson | 86.5 | 68.7 | 83.3 | 89.0 | 88.1 | 89.5 | 86.3 | 93.6 | 93.1 | 71.2 (1.8) | 77.4 (0.9) |
| MaxCos | 88.4 | 69.6 | 82.7 | 88.2 | 89.9 | 92.2 | 89.7 | 96.1 | **98.4** | 92.2 (0.3) | 95.5 (0.4) |
| ReAct | 87.4 | 75.0 | 80.1 | 87.2 | 82.3 | 90.4 | **95.8** | **96.6** | 91.6 | 92.2 (0.3) | 94.5 (0.4) |
| ODIN | 85.4 | 72.9 | 80.3 | 83.9 | 87.7 | 88.8 | 90.0 | 91.4 | 88.3 | 92.2 (0.5) | 93.6 (0.4) |
| DICE | 85.1 | 70.2 | 77.4 | 84.1 | 82.5 | 88.6 | 91.6 | 94.4 | 91.9 | 85.5 (0.3) | 90.1 (0.4) |
| Energy | 85.0 | 72.1 | 79.6 | 83.1 | 87.2 | 88.7 | 90.0 | 90.7 | 88.4 | 91.9 (0.3) | 93.4 (0.4) |
| Igeood | 84.7 | 71.4 | 80.1 | 83.0 | 88.8 | 88.0 | 88.8 | 90.2 | 87.6 | 91.0 (0.3) | 93.3 (0.3) |
| VIM | 84.3 | 66.4 | 78.9 | 80.7 | 89.3 | 90.3 | 83.7 | 87.9 | 97.5 | 92.2 (0.5) | 95.4 (0.4) |
| KL-M | 84.3 | 73.9 | 80.7 | 86.1 | 87.3 | 85.7 | 85.2 | 90.0 | 85.3 | 86.9 (0.6) | 91.4 (0.9) |
| Doctor | 84.2 | 75.9 | 80.6 | 85.1 | 87.0 | 85.1 | 86.7 | 89.7 | 83.8 | 85.2 (0.6) | 89.9 (0.4) |
| RMD | 83.5 | **78.2** | 82.7 | 87.7 | 82.9 | 84.9 | 81.3 | 87.6 | 82.7 | 89.9 (0.3) | 93.1 (0.6) |
| MSP | 83.5 | 75.5 | 79.9 | 84.5 | 86.1 | 84.1 | 85.9 | 88.7 | 83.0 | 83.6 (0.5) | 89.0 (0.4) |
| KNN | 83.4 | 64.3 | 79.6 | 83.3 | 88.0 | 87.2 | 83.0 | 84.1 | 97.6 | 84.6 (0.5) | 89.2 (0.8) |
| GradN | 82.6 | 63.3 | 74.4 | 83.1 | 76.2 | 84.4 | 91.1 | 96.0 | 92.5 | 49.7 (1.0) | 67.4 (1.2) |
| Maha | 69.6 | 55.3 | 65.7 | 70.3 | 70.6 | 73.9 | 60.0 | 72.7 | 88.4 | 71.2 (1.8) | 77.6 (1.8) |

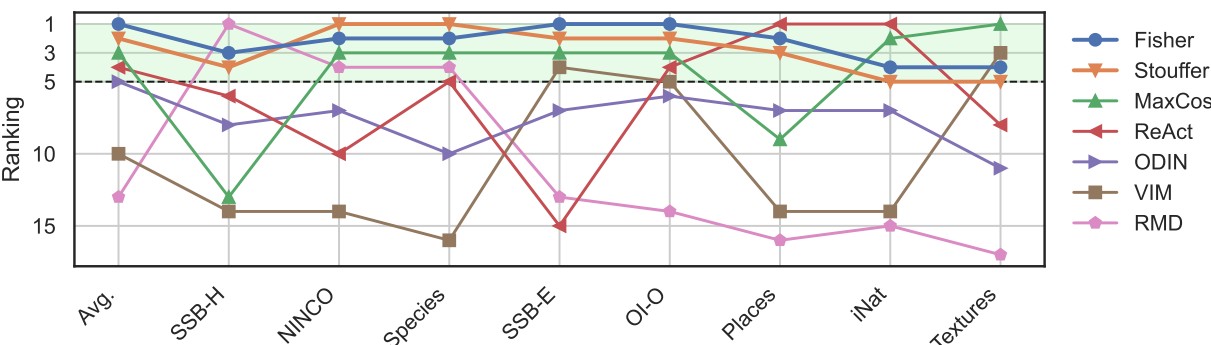

Figure 4: Ranking in terms of AUROC for a few selected methods for the ResNet-50 model. Note that the two displayed methods to combining tests obtain a top-5 ranking in every dataset, while state-of-the-art individual detectors vary significantly in performance.

results displayed in Figure 10, except for the network scale impact, where we obtained more or less the same results for all sizes. On the right-hand side of Table 2, we showed that for both shifts, we demonstrated improved performance by combining p-values, especially with Fisher's method. We also observe from the table that the novelty shift benchmark is slightly easier than the covariate shift benchmark, which is probably biased because most OOD detectors were developed for the novel class scenario. Additional results are available in the Appendix A.

**Results in a sequential stream.** Table 4 displays the average results for the ImageNet-C dataset, including 19 kinds of covariate drifts. We can observe that the most performing methods are the scores function based on the softmax and logit outputs and that Fisher's method is on par with top-performing methods. We emphasize that, even though MSP and Doctor work well in this benchmark, they demonstrated poor

Table 3: Comparative performance in terms of average AUROC for the OOD detection benchmark.

| Normaliz. | Range | Simple Combination | | | Ours | |
|---|---|---|---|---|---|---|
| | | Mean | Min | Max | Stouffer | Fisher |
| Min-Max | $(-\infty, \infty)$ | 89.1 | 87.9 | 85.6 | - | - |
| Standard | $(-\infty, \infty)$ | 89.3 | 87.4 | 86.7 | - | - |
| Quantile | $[0, 1]$ | 89.3 | 88.5 | 86.5 | 89.6 | **89.8** |

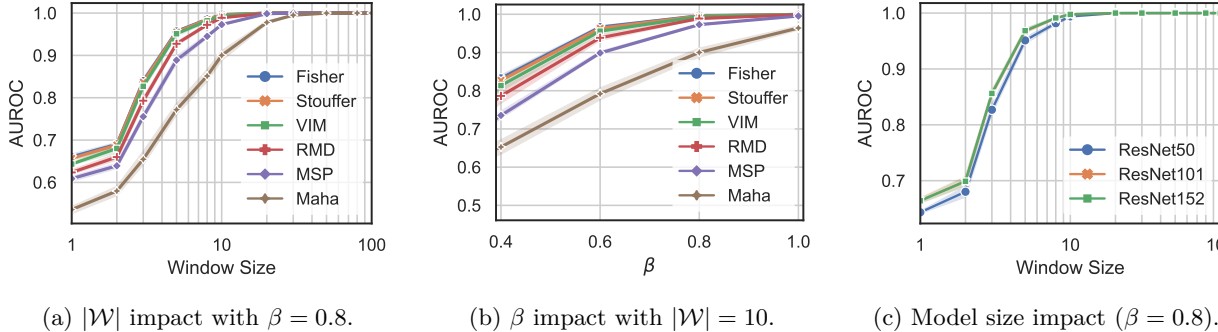

(a) $|\mathcal{W}|$ impact with $\beta = 0.8$.   (b) $\beta$ impact with $|\mathcal{W}| = 10$.   (c) Model size impact ($\beta = 0.8$).

Figure 5: Independent data shift (OpenImage-O) detection performance on a ResNet-50 model (ImageNet).

performance on other benchmarks, notably on Table 2. This supports our claim that combining scores is the most effective approach for improving robustness and performance in general data shift detection.

Table 4: Average Pearson's correlation coefficient with the hidden accuracy with one standard deviation in parenthesis for top and bottom performing detection methods across 19 different corruptions on the sequential data shift detection scenario on a ResNet-50 model.

| | Fisher | Doctor | MSP | Igeood | ... | KNN | RMD | GradN | Maha |
|---|---|---|---|---|---|---|---|---|---|
| Avg. | 0.96 (0.03) | 0.96 (0.03) | 0.96 (0.03) | 0.95 (0.03) | ... | 0.92 (0.07) | 0.92 (0.03) | 0.91 (0.07) | 0.81 (0.21) |

**On the distillation of the best subset of detectors.** We provide a supervised study to showcase the potential impact of finding an optimal subset of detectors. We computed the performance of all possible subsets of $j < k$ methods, and we report our results in Figure 6. We found out that 1.) surprisingly, removing the worse detector from the pool does not necessarily increase performance; 2.) increasing the size of the subset improves probable detection on average and on worst performance; 3.) best subset selection benefits harder to find OOD samples; and 4.) not surprisingly, the best combination for the easy benchmark may be very different from the best subset on the harder one. We also list the best subset of four methods on average performance: {GradN, ReAct, MaxCos, RMD}, on an easy dataset (SSB-Easy): {DICE, MaxCos, KL-M, VIM}, and on a hard dataset (SSB-Hard): {MSP, GradN, ReAct, RMD}. Their AUROC and relative gain w.r.t all methods combined together are equal to 91.4 (+1.8%), 92.0 (+1.1)%, and 79.7 (+4.9%), respectively. *These observations support the main claim of the paper that in a data-free scenario with specialized methods, combining all of them should greatly improve the safety of the underlying system.*

**Limitations.** Our study acknowledges that there is no one-size-fits-all detector or a universally superior combination method, a finding supported by previous research (Heard & Rubin-Delanchy, 2017; Fang et al., 2022). This recognition underlines the inherent complexity of real-world ML applications. Additionally, we recognize that the empirical cumulative distribution function may be susceptible to estimation errors, and the effectiveness of individual detector score functions can influence the performance of the aggregated score. Even though this work stands to improve the reliability of OOD and data-shift detection that allows for safer deployment of machine learning models, there's a risk associated with becoming too confident in the ability of these detection mechanisms. If machine learning practitioners become overly reliant on the OOD detector, they may deploy these models into domains or situations where the detection mechanism fails to

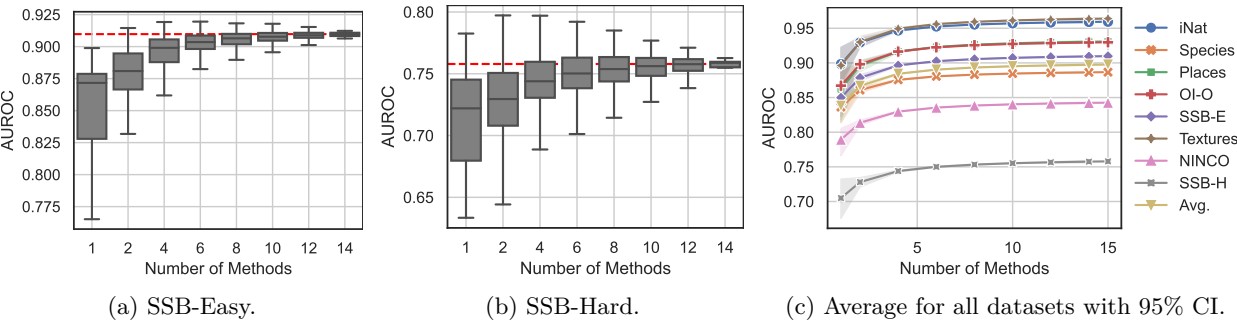

(a) SSB-Easy.  (b) SSB-Hard.  (c) Average for all datasets with 95% CI.

Figure 6: Evaluation of all possible subsets of detectors on the OOD detection benchmark. The dashed red line indicates the performance combining all detectors.

identify OOD data accurately. It is also important to note that although our investigation primarily focused on computer vision applications, similar techniques can be applied to diverse scenarios and application domains.

**Future directions.** Several avenues for future research remain open for exploration. One promising direction involves investigating the performance patterns of detectors across various types of drifts to facilitate subset selection, ultimately improving detection accuracy. However, this may necessitate validation on held-out labeled data or domain expertise to accurately reflect the prior importance of the p-values. Additionally, our proposed algorithm could be integrated into incremental and online learning algorithms, enhancing their adaptability to evolving data streams and offering exciting opportunities for advancing machine learning applications. Furthermore, an intriguing future direction entails designing a method that is instance-dependent, yielding different detector weights for different instances, given our demonstration that various scoring strategies are effective for different types of data inputs.

## 7  Summary and Concluding Remarks

This paper presents a versatile and efficient method for combining detectors to effectively handle shifts in data distributions. By transforming diverse scores into p-values and leveraging meta-analysis techniques, we have illustrated the creation of unified decision boundaries that mitigate the risk of catastrophic failures seen with individual detectors. Our use of Fisher's method, adjusted for correlated p-values, demonstrates strong interpretability as a detection criterion. Through meticulous empirical validation, we've confirmed the effectiveness of our approach in both single-instance out-of-distribution detection and window-based data distribution shift detection, achieving notable robustness and detection performance across diverse domains. Looking forward, our framework establishes a solid groundwork for enhancing the safety of AI systems.

### Acknowledgments

This work has been supported by the project PSPC AIDA: 2019-PSPC-09 funded by BPI-France and was granted access to the HPC/AI resources of IDRIS under the allocation 2023 - AD011012803R2 made by GENCI.

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

# A    Appendix

## A.1    Combining Multiple P-Values with Stouffer's method

The Stouffer et al. (1949) test statistics for combining p-values is given by:

$$s_S(\cdot) = \sum_{i=1}^{k} \Phi^{-1}(\mathrm{p}_i(\cdot)) \tag{12}$$

where $\Phi^{-1}$ is the *probit*, i.e., $\Phi^{-1}(\alpha) = \sqrt{2}\,\mathrm{erf}^{-1}(2\alpha - 1)$, where erf is the Gauss error function. If the p-values are independent, $s_S(\cdot) \sim \mathcal{N}(0,1)$, where $\mathcal{N}(\mu, \sigma^2)$ is the normal distribution with mean $\mu$ and standard deviation $\sigma$.

## A.2    Correcting for correlated p-values with Hartung's method

Hartung (1999) method aims to correct Stouffer's test for correlated p-values. The group statistics write:

$$s_H(\cdot; \boldsymbol{w}, \rho) = \frac{\sum_{i=1}^{k} w_i \Phi^{-1}(\mathrm{p}_i(\cdot))}{\sqrt{(1-\rho)\sum_{i=1}^{k} w_i^2 + \rho\left(\sum_{i=1}^{k} w_i\right)^2}} \underset{H_0}{\sim} \mathcal{N}(0,1) \tag{13}$$

with $\rho$ a real-valued parameter and $\sum_{i=1}^{k} w_i \neq 0$. Hartung showed that an unbiased estimator of $\rho$ based on $\mathrm{p}_i$ under $H_0$ is given by:

$$\hat{\rho} = 1 - \mathbb{E}\left[\frac{1}{k-1}\sum_{i=1}^{k}\left(\Phi^{-1}(\mathrm{p}_i) - \frac{1}{k}\sum_{i=1}^{k}\Phi^{-1}(\mathrm{p}_i)\right)^2\right]. \tag{14}$$

Assuming equal weights, we repeated a similar experiment as the one of Figure 1, replacing the chi-squared with a standard normal to see how well the correction works. We can observe in Figure 7 that the corrected statistic indeed approximates a standard normal distribution. Unlike Brown's method, Hartung's method corrects the statistics directly instead of correcting the parameters of the underlying distribution.

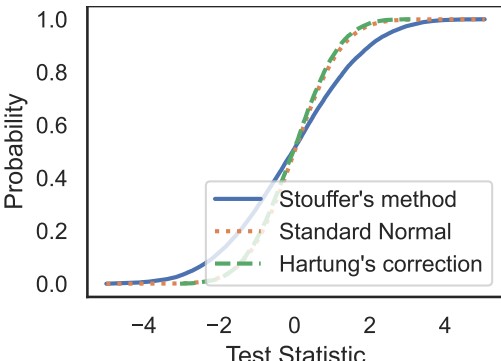

Figure 7: Stouffer's method corrected for correlated p-values with Hartung's method to obtain a standard normal distribution when evaluated on in-distribution data (null hypothesis), also obtaining interpretable results.

## A.3 Additional Plots

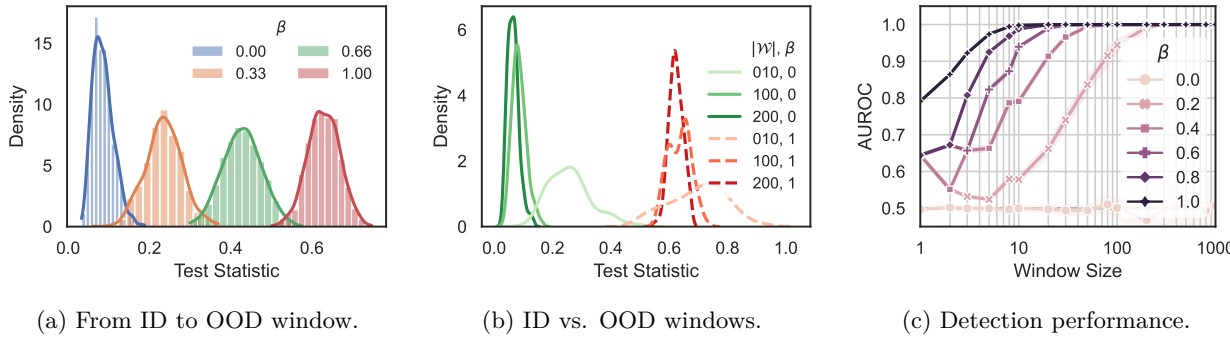

(a) From ID to OOD window.

(b) ID vs. OOD windows.

(c) Detection performance.

Figure 8: Test statistic behavior and detection performance in function of the covariate shift intensity and window size. Experiments ran on a ResNet-50.

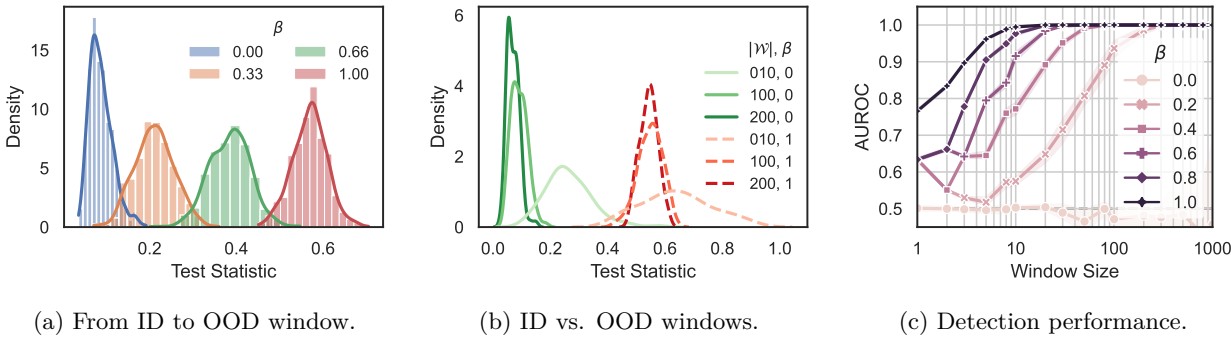

(a) From ID to OOD window.

(b) ID vs. OOD windows.

(c) Detection performance.

Figure 9: Test statistic behavior and detection performance in function of the covariate shift intensity and window size. Experiments ran on a ViT-L-16.

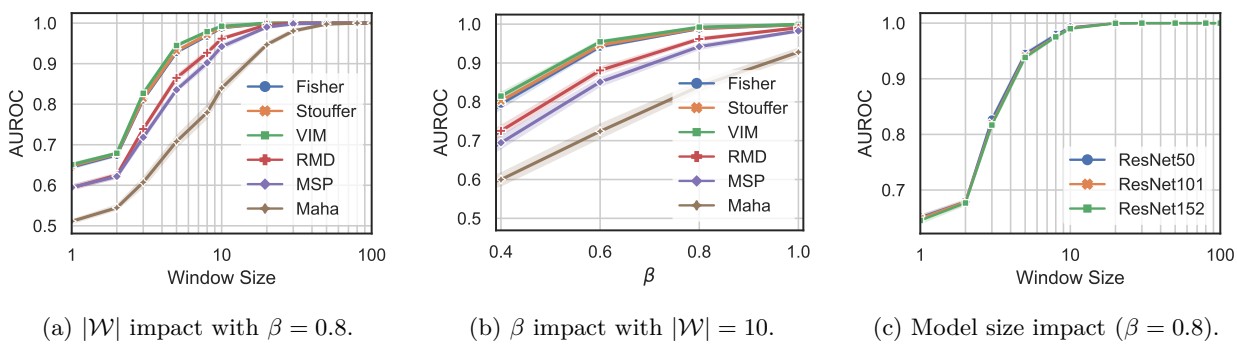

(a) $|\mathcal{W}|$ impact with $\beta = 0.8$.

(b) $\beta$ impact with $|\mathcal{W}| = 10$.

(c) Model size impact ($\beta = 0.8$).

Figure 10: Covariate shift (ImageNet-R) detection performance on a ResNet-50 model (ImageNet).

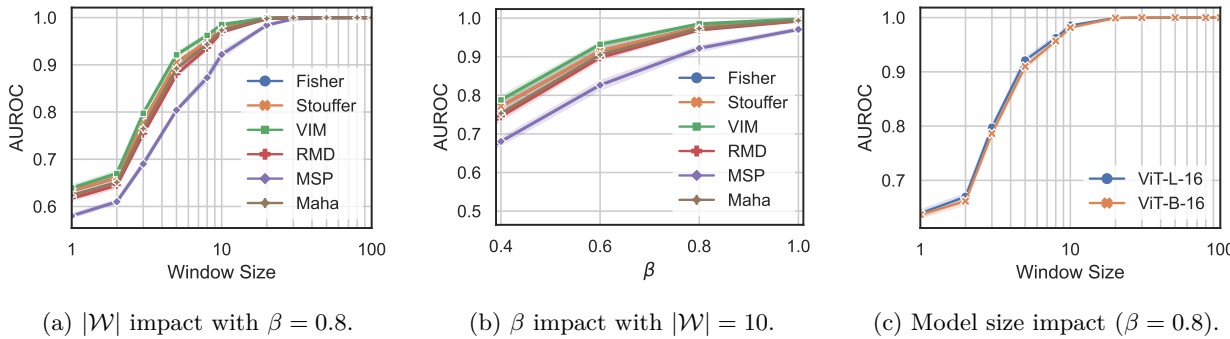

(a) $|\mathcal{W}|$ impact with $\beta = 0.8$.  (b) $\beta$ impact with $|\mathcal{W}| = 10$.  (c) Model size impact $(\beta = 0.8)$.

Figure 11: Covariate shift (ImageNet-R) detection performance on a ViT-L-16 model (ImageNet).

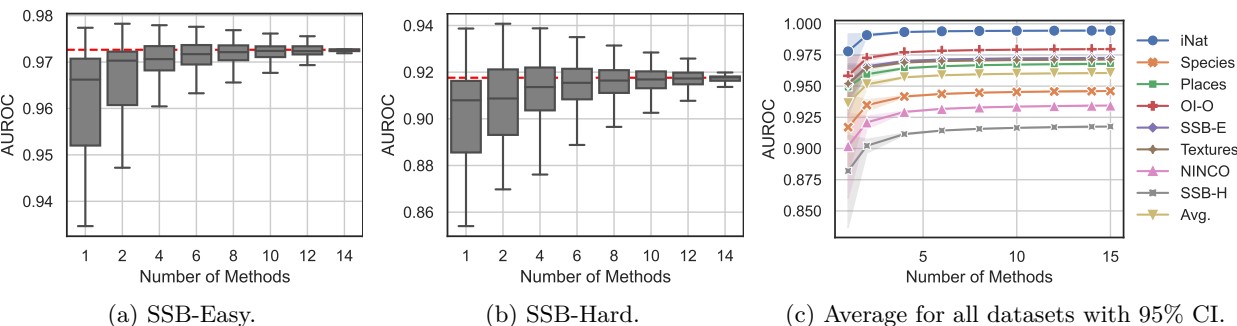

(a) SSB-Easy.  (b) SSB-Hard.  (c) Average for all datasets with 95% CI.

Figure 12: Evaluation of all possible subsets of detectors on the OOD detection benchmark for a ViT-L-16 model. The dashed red line indicates the performance combining all detectors.

## A.4  Additional Tables

Table 5: Numerical results in terms of AUROC (values in percentage) comparing p-value combination methods against literature for a ViT-L-16 model trained on ImageNet.

| Method | Avg. | SSB-H | NINCO | Spec. | SSB-E | OI-O | Places | iNat. | Text. |
|---|---|---|---|---|---|---|---|---|---|
| Maha | **96.8** | 92.7 | **94.8** | 96.6 | 97.4 | **98.6** | 96.9 | **99.8** | 97.6 |
| VIM | 96.6 | 92.1 | 93.9 | 95.6 | **97.7** | 98.5 | 96.7 | 99.7 | **98.2** |
| RMD | 96.1 | 92.4 | **94.8** | 96.2 | 96.3 | 97.9 | 95.7 | 99.5 | 95.6 |
| Fisher/Brown | 96.1 | 91.8 | 93.4 | 94.6 | 97.3 | 98.0 | 96.8 | 99.5 | 97.1 |
| Vovk | 96.1 | 91.8 | 93.4 | 94.6 | 97.3 | 98.0 | 96.8 | 99.5 | 97.1 |
| Simes | 96.0 | 91.7 | 93.4 | 94.6 | 97.1 | 98.0 | **97.0** | 99.5 | 97.0 |
| Stouffer/Hartung | 96.0 | 91.5 | 93.3 | 94.4 | 97.3 | 97.9 | 96.7 | 99.4 | 97.1 |
| ReAct | 95.9 | **93.9** | 94.7 | **96.9** | 96.6 | 97.8 | 91.1 | 99.5 | 96.3 |
| Edgington | 95.7 | 90.9 | 92.8 | 93.9 | 97.1 | 97.7 | 96.8 | 99.2 | 97.1 |
| Energy | 95.6 | 91.0 | 92.5 | 93.2 | 97.3 | 97.8 | 96.4 | 99.3 | 97.1 |
| Tippet | 95.5 | 90.9 | 92.3 | 94.6 | 96.4 | 97.6 | 96.9 | 99.3 | 96.2 |
| Pearson | 95.5 | 90.4 | 92.4 | 93.6 | 97.1 | 97.6 | 96.8 | 99.0 | 97.0 |
| MaxL | 95.5 | 91.2 | 92.6 | 93.2 | 97.0 | 97.6 | 96.1 | 99.3 | 96.8 |
| ODIN | 95.5 | 91.2 | 92.6 | 93.2 | 97.0 | 97.6 | 96.1 | 99.3 | 96.8 |
| Igeood | 95.4 | 90.8 | 92.6 | 93.2 | 97.1 | 97.6 | 96.0 | 99.2 | 96.7 |
| MaxCos | 94.9 | 89.7 | 91.2 | 92.9 | 97.0 | 96.9 | 96.2 | 98.2 | 97.1 |
| GradN | 94.9 | 90.1 | 91.4 | 91.8 | 96.6 | 97.3 | 96.1 | 99.2 | 96.3 |
| KNN | 93.4 | 85.4 | 89.2 | 91.9 | 96.3 | 96.1 | 94.3 | 97.6 | 96.4 |
| Doctor | 93.1 | 88.9 | 90.3 | 91.8 | 94.1 | 94.8 | 93.2 | 98.4 | 93.7 |
| MSP | 92.5 | 88.2 | 89.5 | 91.3 | 93.5 | 94.0 | 92.4 | 98.0 | 93.0 |
| KL-M | 92.1 | 85.4 | 89.0 | 90.6 | 93.5 | 94.2 | 92.5 | 98.0 | 93.7 |
| Wilkinson | 91.2 | 81.6 | 85.0 | 87.1 | 94.2 | 94.7 | 96.3 | 95.7 | 95.2 |
| DICE | 76.3 | 60.2 | 63.6 | 67.0 | 79.8 | 80.8 | 94.3 | 81.9 | 82.5 |

