# OpenReview forum: "Combine and Conquer: A Meta-Analysis on Data Shift and Out-of-Distribution Detection"
_TMLR — Accepted by TMLR_

### Review · Reviewer_fkys · 2024-03-14

**Summary Of Contributions:**

This paper introduces a method to improve out-of-distribution (OOD) detection by integrating various detection scores that reflect the self-confidence of deep learning models and anomalous behavior in latent space. Traditional combination techniques using simple statistics are insufficient for this task. The solution involves quantile normalization of detection scores into p-values, effectively turning the problem into a multi-variate hypothesis test. This approach allows for the use of meta-analysis tools to create a unified, effective OOD detector with solid decision boundaries. Empirical studies show significant enhancements in robustness and performance across different OOD scenarios, demonstrating the framework's extensibility and marking a first in combining decision boundaries for OOD detection in this manner.

**Audience:**

Yes

**Broader Impact Concerns:**

No concerns.

**Claims And Evidence:**

Yes

**Requested Changes:**

It would be great if the authors can address the questions and weaknesses in the above section.

**Strengths And Weaknesses:**

Strengths:
- The paper tackles an important OOD detection problem, which is of great relevance to real world applications.
- The paper is well written in general.
- The proposed method is technically sound.
- The empirical evaluation is thorough.
- I particularly like the experiments and discussion on window based detection, which is often neglected in typical OOD detection papers.

Weaknesses/Questions:
- As a crucial step, quantile normalization needs to be performed with training/validation data. Typically, both of these dataset do not contain any OOD samples. Moreover, the model could be heavily overfit to the training data. Are there any risks in doing quantile normalization using such datasets? Would this lead to false positives?
- How is the significant level selected?
- Any theoretical guarantee  on the proposed detection function (equation 11)?
- For window-based detection, typically window size can be adjusted. I wonder if there is a way to select an optimal window size?

---

> ### Author Response · Authors · 2024-04-14
>
> We thank the reviewer for the valuable feedback on our manuscript and acknowledge that our paper is well-written, the method is technically sound, and the empirical evaluation is extensive. In the following, we hope to address all of the reviewer's questions:
>
> 1. The quantile normalization is done only with only in-distribution data as a means of testing against the null hypothesis. In our experiments, we used a held-out validation set with 1000 samples. To address the reviewer's question, we also ran experiments by doing the quantile normalization with 1000 randomly selected samples from the training dataset. The results are available in the table below. We can observe a slight decrease in performance in OOD detection in terms of average AUROC overall, indicating a potential increase in false positives for a random threshold. Anyhow, the method exhibits better performance than the single detectors.
>
> | Method 	| Training | Validation |
> |------------|----------|------------|
> | Fisher 	| 89.3 	| 89.8   	|
> | Stouffer   | 89.2 	| 89.6   	|
> | Edgington  | 88.7 	| 89.3   	|
> | Pearson	| 88.7 	| 89.2   	|
> | Simes  	| 88.9 	| 89.2   	|
> | Tippet 	| 88.4 	| 88.5   	|
> | Wilkinson  | 86.2 	| 86.5   	|
>
> 2. The significance level $\alpha$ should be selected to control the detector's false negative rate (FNR). For example, for a desired FNR of 5% or 95% TPR, we would simply set $\alpha=0.05$ as they coincide in our framework (up to sample-size error) with proper quantile normalization and correlation correction (Eq. 11).
> 3. One argument for the lack of theoretical justification for aggregation methods in out-of-distribution (OOD) detection is the complexity of deep architectures and variability of real-world datasets which are characterized by high dimensionality and non-linear relationships among features. OOD detection deals with scenarios where the distribution of unseen data is not explicitly known or well-defined. Additionally, our aggregation method relies implicitly on the training/optimization procedure (e.g., the way that network parameters capture the statistics and information about the training distribution) and the underlying deep neural network architecture, making it challenging to devise a single theoretical framework that adequately captures these important aspects. Traditional theoretical frameworks (e.g., based on measure concentration inequalities using structural properties of the loss function) may struggle to capture the intricacies of such complex data structures, limiting their applicability to aggregation methods for OOD detection. In summary, the lack of theoretical justification for aggregation methods in OOD detection can be attributed to the inherent complexity, high dimensionality and non-linear relationships among features, and uncertainty associated with the understanding of the true distribution of out-of-distribution samples.
> 4. Based on our experiments, individuals aiming to select window sizes for achieving a desired level of detection performance would require prior knowledge of $\beta$ and access to samples from $\nu$. Consequently, to the best of our understanding, no distribution-free optimal method exists for choosing the window size. Nevertheless, we can offer general recommendations based on empirical observations. Our findings (illustrated in Fig. 2.c) indicate that the magnitude of the shift directly influences the required window size for maintaining a consistent desired performance level. For instance, to achieve an AUROC level of 0.9, we would recommend window sizes of [2, 4, 7, 16, 60] for corresponding mixture values (representing the proportion of out-of-distribution data) of [1.0, 0.8, 0.6, 0.4, 0.2] on the ImageNet vs OpenImages mixture benchmark.

---

### Review · Reviewer_gsyy · 2024-03-20

**Summary Of Contributions:**

This paper proposes that for detecting out-of-distribution samples as well as for detecting data shifts, it is beneficial to combine multiple different individual metrics. However, the paper argues that combining such different metrics is not trivial as standard approaches for doing so have issues. Instead, the paper proposes to use the statistical approach of meta-analysis to build a framework for combining individual metrics in a principled way. In a series of experiments, it is convincingly demonstrated that the proposed combination of metrics performs better than each of the respective individual metrics.

**Audience:**

Yes

**Broader Impact Concerns:**

No concerns in this regard.

**Claims And Evidence:**

No

**Requested Changes:**

The main changes I encourage the authors to make is addressing the three main issues (labeled [1], [2] and [3]) that are raised under “Strengths and Weaknesses”.

**Strengths And Weaknesses:**

The main proposal of this paper, i.e. combining multiple individual metrics for out-of-distribution / data shift detection together, is straightforward and clear, and the arguments that are provided for doing so are sensible. The statistical framework based on meta-analysis that is proposed is also well-motivated and reasonably clearly described. I was not able to find any major mistakes in either the proposed framework or the statistical foundations for it. The empirical results seem to convincingly indicate that the proposed approach for combining individual metrics performs better and more consistently than each of the respective individual metrics.

I need to indicate that I am not an expert in out-of-distribution / data-shift detection, so it is difficult for me to properly judge the scientific contribution of the paper in this regard. I am also not an expert in statistical meta-analysis, so it is also hard for me to judge the quality/standard of the paper in this respect.

I found several issues that I think should be addressed before this paper could be accepted for publication.

[1] I think there is some confusion / mix up with regards to the terms *concept drift*, *covariate shift* and *prior shift*. Concept drift is defined as a change in the conditional distribution of Y given X (and with no change in the input distribution X). In subsection 3.3, it is stated that the instances’ class labels (i.e., the Y’s) are not available for the detection of data shifts. It seems to follow from this that it becomes impossible to detect concept drifts, yet the authors still claim to do this (e.g., subsection 5.2). It seems to me that what is described in subsection 5.2 as being a covariate shift, is actually more akin to a prior shift.

[2] I found section 5 hard to read and understand. A reason for this is that I am unfamiliar with most of the datasets / benchmarks that are used, while at the moment it seems the paper is written in such a way that knowledge of these is assumed. I would encourage the authors to make changes to this section so that it is also possible for readers unfamiliar with the used datasets to follow it.

[3] In subsection 4.1 several conceptual arguments are made for why combining different individual metrics in standard or naïve ways is not optimal. However, I could not find an empirical evaluation of these standard approaches. I think it would be important to include these standard approaches in the empirical comparisons, to back up the conceptual arguments from subsection 4.1.

Minor issues:
- several issues with formatting of references (e.g., top of last paragraph p2)
- inconsistent use of capitalisation (e.g., title of subsections 4.3 and 4.4)

---

> ### Author Response · Authors · 2024-04-14
>
> We thank the reviewer for their thorough review of our paper and for recognizing that our method is presented clearly and well-motivated. In the following, we hope to address the reviewer's concerns in the best way possible:
>
> [1] Section 5.2 explains how we simulate concept drift by introducing novelty w.r.t ImageNet as new concepts from natural images. This could be seen as a concept shift as new labels are introduced to $\mathcal{Y}$, and the inputs are still natural images. Note that, as decomposed in eq. 2, it is natural that a change in $Y|X$ results in a change in $Y$. Section 3.3 states that the labels are unavailable at test time for the detection scores, which only have access to the NN model $f$ and the input $x$. A practitioner can only stratify the kind of drift that actually happened if the test set is labeled a posteriori. Consequently, the paper's primary focus lies in detecting data drift in a broad sense rather than explicitly delving into the mechanisms behind the drift. We added a statement to the main manuscript to clarify this point.
>
> [2] We thank the reviewer for their feedback on the level of detail. We made several modifications to section 5 (in blue in the updated manuscript) to clarify the different kinds of datasets used and better detail the experimental setup, especially for the data drift scenarios.
>
> [3] To address this, we ran experiments with simple normalization statistics (Min-Max scaling and Standard Normalization) combined with simple score combinations (mean, min, max). We compared them with the quantile normalization and the proposed score combination methodology. The results in terms of average AUROC in the OOD detection benchmark are available in the table below. We also added this table to Section 6 (Table 3 in the updated manuscript).
>
> |           	|                	| Simple Combination |           	|          	| Ours         	|          	|
> | ------------- | ------------------ | ------------------ | ------------- | ------------ | ---------------- | ------------ |
> | Normalization | Range   | Mean     	| Min       | Max      | Stouffer/Hartung | Fisher/Brown |
> | Min Max   	| $(-\infty,\infty)$ | 89.1    	| 87.9   | 85.6  | \-        | \-      |
> | Standard  	| $(-\infty,\infty)$ | 89.4     	| 87.4   | 86.7  | \-        | \-      |
> | Quantile  	| $[0,1]$        	   | 89.3     	| 88.6   | 86.5  | 89.6   | 89.8 |
>
>
> We also thank the reviewer for spotting minor formatting issues, which we have corrected in the revised version of the manuscript.

---

> > ### Comment · Reviewer_gsyy · 2024-04-19
> > **Response to rebuttal**
> >
> > Thanks for the rebuttal. I'm mostly satisfied with the answers, but I am still uncertain about the concept shift that the authors introduce in section 5.2. In section 3.2, concept shift is defined as a change in Q(Y|X) while q(X) remains constant. This means that the input distribution (i.e., q(X)) stays the same, but that for some inputs the label changes. For example, an image that was initially labeled as "cow" might now be labelled as "grass" (i.e., the image stays the same, only the label assigned to that image changes). Is this indeed what is being done in section 5.2?
> >
> > (I'm afraid that I am unfamiliar with the dataset OpenImage-O, which makes it hard for me to really understand how this experiment is set up. Perhaps it could help to show a visualization of the datashift using example images?)

---

> > > ### Author Response · Authors · 2024-04-20
> > >
> > > We appreciate the reviewer's attention to our rebuttal.
> > >
> > > To illustrate the concept drift better, consider an example of developing a computer vision model to classify any object encountered in daily life. A team collects and labels 1.3 million natural images from human interaction, identifying 1000 concepts. They release a preliminary model, which, when shown a photo of a donut, classifies it as a bagel. The input is still a natural image, but the label space limitation causes a novelty encounter, as the model hasn't learned the distinction between bagels and donuts. This scenario is simulated in Section 5.2 by showing images from the OpenImage-O dataset to a model trained on ImageNet.
> > >
> > > I refer the reviewer to this [Github link](https://github.com/j-cb/NINCO?tab=readme-ov-file#examples-images-from-some-of-the-most-difficult-ninco-ood-classes) where the authors compared the most difficult OOD images by comparing them to its closest correspondence in ImageNet. Please let us know if this responds to your question.

---

> ### Comment · Reviewer_gsyy · 2024-04-20
> **Request for further clarification**
>
> Thank you for the fast response and the clarification. It is a bit clearer to me, but still not completely.
>
> To continue with the example of donuts and bagels, I can think of two possible options.
>
> OPTION 1
> \
> In the initial dataset (ImageNet in this case, I believe) there are images of both donuts and bagels, but both are labelled as 'bagels'. In the new dataset (OpenImage-O in this case, I believe), there is the same distribution of images of both donuts and bagels, but now there is a more fine-grained labelling which includes both 'bagels' and 'donuts'.
>
> OPTION 2
> \
> The initial dataset only contains images of bagels, not of donuts. But because donuts look like bagels, the model that is learned classifies donuts as bagels. The new dataset contains images of donuts.
>
> Which of these two options corresponds to the experiment in section 5.2?

---

> > ### Author Response · Authors · 2024-04-21
> >
> > Thank you for following up on this. The experiments outlined in Section 5.2, Paragraph 1, align with Option 2. The initial dataset is indeed ImageNet, and the new (test) dataset contains images of bagels and donuts as it is a mixture of ImageNet and OpenImage-O.

---

> ### Comment · Reviewer_gsyy · 2024-04-21
> **Concept shift or prior shift**
>
> Thank you again for the quick response.
>
> If I understand it correctly, this means that by going from the original dataset (ImageNet) to the new dataset (ImageNet + OpenImage-O), there is an addition of images from a new class (in this case 'donuts'). That is, in original dataset there were no images of donuts, but in the new dataset there are. This thus means that there is a change in the input distribution q(X). This means that this distribution shift is not a concept shift, because a concept shift is defined as a change in Q(Y|X) while q(X) remains constant (see top of p4).
>
> Rather, it seems to me that the distribution shift described above is a prior shift: in the initial dataset (ImageNet) there are only bagels, in the new dataset (ImageNet + OpenImage-O) there are bagels and donuts.
>
> I understand that this is not at the core of the paper, and this issue does not necessarily invalidate the paper's main claims, but I think it is important to make sure this is correct.

---

> > ### Author Response · Authors · 2024-04-22
> > **Improved clarity with novelty drift and relaxing distributional constraints**
> >
> > We appreciate the reviewer's attention to detail and for improving the clarity of our experimental setup.
> >
> > The authors acknowledge and agree that the paper's primary focus isn't on the taxonomy of these concepts. In our efforts to make our work more accessible, we've decided to follow the taxonomy provided in [1] and use the terms "novelty drift" or "concept evolution" instead of "concept drift." We also relaxed the distributional constraints for this kind of drift. We have included these modifications where appropriate in the updated version of our manuscript. We kindly ask the reviewer to review the top of p. 4 and Section 5.2, par. 1.
> >
> >
> > ### References:
> > [1] Masud, M., Gao, J., Khan, L., Han, J., & Thuraisingham, B. M. (2011). Classification and novel class detection in concept-drifting data streams under time constraints. IEEE Transactions on Knowledge and Data Engineering, 23(6), 859-874. Article 5453372.

---

> > > ### Comment · Reviewer_gsyy · 2024-04-23
> > >
> > > Thank you again for the quick response.
> > >
> > > I think these changes address the most pressing issue. I have to say that I still do not understand why the drift in Section 5.2, par 1, is not a prior shift (or indeed, with the new terminology, what the difference is between a novelty drift and a prior drift), but I can no longer point towards clear mistakes. Given that this taxonomy is not the main contribution of the paper, I think this is probably acceptable. However, I would encourage the authors to indicate in the manuscript that their taxonomy of different kind of drifts is not the main point of the paper, and that this taxonomy is possibly somewhat "approximate".

---

> > > > ### Author Response · Authors · 2024-04-24
> > > >
> > > > We thank the reviewer for their feedback. Next, we explain why we decided to go with novelty drift and not prior drift and the added statement to the paper to clear any misunderstanding.
> > > >
> > > > A prior probability shift also covers situations where the test set is skewed or imbalanced, and there's no novelty. We want to emphasize that we intentionally injected novelty into the test data, thus novelty drift.
> > > > We carefully introduced the following statement in Section 3.2 to improve our paper: "However, this list is not exhaustive as the principal objective of this paper does not lie in the precise categorization of diverse drift phenomena but rather in the establishment of a more robust detection framework for distinct scenarios."

---

### Review · Reviewer_FtKf · 2024-03-31

**Summary Of Contributions:**

This paper presents a method to combine out-of-distribution or domain-shift detectors, using techniques from "classical" statistics.  Scores for each detector are calibrated using quantile p-values, then combined using $\chi^2$ or other combination method.  These are also monotonically adjusted for correlative dependence, with a distribution scaling.  Numerous experiments using imagenet demonstrate the effectiveness of the method and its robustness.

**Audience:**

Yes

**Broader Impact Concerns:**

This work stands to improve reliability of OOD and domain-shift detection, which can enable safer application of models.  However, over-confidence in an OOD detection ability may also facilitate wider deployments of classifiers, including into domains where the OOD detection itself fails.  This could be discussed in the Limitations section.

**Claims And Evidence:**

Yes

**Requested Changes:**

None required.  I mentioned a few in my review above that I feel would strengthen the paper.

**Strengths And Weaknesses:**

Overall, this is a simple and well-described method, using statistics to combine existing detectors in a way that results in well-behaved (nicely varying between 0 and 1) OOD/domain-shift score functions, that outperform any individual detector, and most combinations of fewer detectors as well.

Most reassuringly, the variance in performance decreases substantially as more methods are used, and this performance level tends towards the higher available for each dataset, as shown in figs 4 and 6.

The text mentions that some distribution shifts may not have an impact on model performance, but I didn't see anything that measures how much this method might improve (or worsen) detection of these cases where it might not be desired.  For example, if a model is trained on imagenet-1k, but then applied to a subset of only 10 (in-distribution) classes, will this detect a window-based distribution shift?  This may or may not be desired depending on the application.

In addition, I would have liked even more on the effectiveness of the approach as different detectors are included or omitted from the system.  p.11 presents a good analysis of this, showing the performance of all possible selections.  But I still would have been in explorations such as, for example, if one selects the top 5 methods according to one subset of measurement datasets (or synthetically corrupted val set), how well does that work compared to selecting all of them, for the held-out subset of datasets?  Likewise, I'd be curious to see how well combining all but one or two of the best overall methods (maxcos) compares to using just that one (or two) method(s)?


More minor questions/comments:

- The text mentions several combination methods, including a few simple types of average, but I didn't initially see these compared because they are referred to only by the names from 4.3 in table 2.  A reminder of which is which (i.e. using the same defs as in 4.3) would be good to include in the table row or caption.

- Is there any binning or smoothing of p-values?

- eq 6:  lowercase $w$ is not defined.  what is the sup_$w$ computed over?  (it appears this is defined later on in eq 7 as the empirical cdf threshold)


- Fig 5c:  Model size having only a small effect is a nice behavior.  How much of this is from the ensembling inherent in the score combination method?  What if fewer (or only one) shift detection method are used --- does this difference get any larger?  If so, this would be another benefit of combining methods (less dependence on model size/choice).


- what is an example of virtual drift that doesn't affect the model, related to the decompositions in eq (3), as was described for three types affecting shifts?

- could use a little more explicit explanation of why p values are modeled as uniform RVs.  This is by construction of the p values, I believe.

- similarly, could elaborate more on non-independence, though this is pretty clear:  an in-distrib (high score) point is likely to be scored high for any useful score function, so if one function has high score, a second function is likely too as well

- eq.2:  this linear mixture is not used in this method, only in evaluations.  might make more sense to define this in the evaluation methods procedure instead, as mentioning it earlier sets the stage for trying to recover the individual components $p, v, \beta$ which is not actually the method described

- also note: I did not check the citations of all the referenced theorems or lemmas for correctness of application, but did consider whether the conclusions they are used to support seemed reasonable according to the authors' descriptions, and found these descriptions clear.

---

> ### Author Response · Authors · 2024-04-14
>
> We thank the reviewer for their valuable feedback and highlighting that our method has significantly reduced the variance in detection performance. We have incorporated the recommendations suggested by the reviewer in our revised manuscript as follows:
>
> 1. We updated the caption of Table 2 accordingly.
> 2. No, we did not apply binning or smoothing of p-values.
> 3. We updated eq. 6 to make it fully compatible with eq. 7.
> 4. A single detector exhibits similar average performance on different model sizes but with a higher variance w.r.t combining multiple detectors.
> 5. An example of virtual drift that doesn’t affect the model’s performance is when the conditional P(Y|X) remains unchanged. For instance, the covariate distribution P(X) changes (e.g., mild corruption to an image sensor) without affecting the performance of the classifier.
> 6. The p-value is a uniformly distributed r.v. if the null hypothesis is true and for continuous test statistics. This comes from the probability integral transform. We refer the reviewer to Fisher, R.A. Statistical methods for research workers. 1925 for more details.
> 7. We appreciate the reviewer for raising this insightful point. As pointed out by the reviewer, an in-distribution data point (one with a high score) is likely to receive high scores across various useful score functions. Naturally, if the underlying distributions between the two hypothesis tests were provided, then the optimal score function (as per the Newman-Pearson test) could be employed without the need to incorporate any additional scores. However, given that the underlying distributions (in-distribution and OOD) are unknown and only partial information (derived from the training samples) is available about one of the hypotheses, each score may carry relevant information for the decision. This is because each score induces a distinct probability mass transformation of the same data point, contingent upon the underlying hypothesis under test. Viewed in this light, this multi-score approach can be seen as a means of enhancing the diversity of the scores by aggregating different scoring mechanisms, which is helpful for testing different out-of-distribution scenarios. Consequently, this increases the likelihood that at least one will successfully identify the correct hypothesis. It is very much the same principle as MIMO (multiple antennas) in communication systems, which increases the diversity of the observations at the receiver and, thus, the capacity of the receiver to decode the message. We have included a clarification in the revised version of the manuscript to address this concern.
> 8. The purpose of eq. 2 is to characterize the magnitude of the data drift, serving as a foundational equation to effectively articulate the problem of detecting probability shifts. We acknowledge the concerns raised by the reviewer regarding potential confusion stemming from this equation. In response, we have included a clarifying note, elucidating our focus on "detecting instances of data shift occurring between the training and testing distributions, rather than attempting to estimate the mixing parameter $\beta$ or the true pdfs involved."
>
> In the following, we endeavor to address additional inquiries posed by the reviewer to the best of our ability:
>
> 1. "If a model (Imagenet-1k) is applied to a subset of samples from 10 (in-dist.) classes, will this detect a window-based dist. shift?"
> In the independent window-based data drift detection pipeline, this example of a prior distribution shift is imperceptible by the detector. However, for the dependent window-based data drift detection, if the **same** 10 classes appear in multiple consecutive windows, the score may detect a slight change in the cumulative statistics, indicating a drift.
> 2. "If one selects the top 5 methods according to one held-out dataset, how well does that work compared to selecting all of them?"
> The response to this question hinges on the specific subset of corrupted samples withheld for evaluation. Examining the upper ends of the whiskers in the box plots depicted in Fig. 6a and 6b, we observe that the performance gap between the top 4-6 detectors for SSB-Easy, compared to aggregating all available detectors, is approximately +1% within the same dataset. However, for SSB-Hard, this discrepancy widens to around +4%. It's noteworthy that the composition of the top 4-6 detectors varies between these datasets, implying that extrapolating the performance of this validation procedure to multiple out-of-distribution datasets is inherently constraining.
> 3. "How does combining all but the best overall method compare to just this one?"
> We conducted an experiment by excluding MaxCos from the pool of aggregated detectors. The avg. AUROC achieved using Fisher statistics was 89.5, whereas MaxCos individually achieved an avg. AUROC of 88.4.
>
> We also incorporated a statement related to the broader impact concerns raised by the reviewer on the limitations section.

---

### Decision · Action_Editor_6Fvh · 2024-05-28

**Recommendation:** Accept as is

**Comment:**

This paper presents a method for combining multiple out of distribution detection methods / scores in order to obtain a more robust combined score.  The authors argue that very simple methods for combining scores does not work well, so instead they develop a more elaborate strategy using p-values in the context of a statistical hypothesis test.

The reviewers all voted to accept the paper (two leaning accept and one accept. The reviewers found the paper well motivated, the methodological contribution mathematically sound and well justified, and they found the empirical results convincing.  One reviewer remarked that additional ablations would strengthen the work ("I would have liked even more on the effectiveness of the approach as different detectors are included or omitted from the system").  Although the reviewers have expressed that additional experiments / ablations would make the work stronger, they found the existing empirical work convincing and thorough enough.  Therefore, we leave it up to the authors to determine whether they would like to add these, but won't require revisions for acceptance.

One reviewer found that, although the authors addressed their concerns in the author response, they didn't find the response particularly helpful or insightful. They cited this as a reason to 'lean accept' rather than outright accept.

**Audience:**

The paper is on out of distribution detection for deep learning models, which is both a major subfield of active research and a major practical issue for practitioners.

**Claims And Evidence:**

The reviewers all found that the claims made in the paper are supported by the empirical evidence.  In particular the paper claims to improve on individual OOD metrics by combining metrics, and shows that naive strategies don't work as well.  The reviewers found the experiments convincing of these claims.